# Prevalence of yaws and syphilis in the Ashanti region of Ghana and occurrence of H. ducreyi, herpes simplex virus 1 and herpes simplex virus 2 in skin lesions associated with treponematoses

Yaw Agyekum Boaitey[1,2], Alex Owusu-Ofori[2,3], Amarachukwu Anyogu[4], Farhang Aghakhanian[5], Natasha Arora[6], Jonathan B. Parr[5], Philipp P. Bosshard[7], Saki Raheem[1], Pascale Gerbault[8]*

1 School of Life Sciences, University of Westminster, London, United Kingdom, 2 Komfo Anokye Teaching Hospital, Kumasi, Ghana, 3 Kwame Nkrumah University of Science and Technology, Kumasi, Ghana, 4 School of Biomedical Sciences, University of West London, London, United Kingdom, 5 University of North Carolina at Chapel Hill, Chapel Hill, North Carolina, United States of America, 6 Zurich Institute of Forensic Medicine, University of Zurich, Zurich, Switzerland, 7 University Hospital Zurich, University of Zurich, Zurich, Switzerland, 8 Department of Genetic and Evolution, Laboratory of Anthropology, Genetic and Population, University of Geneva, Genève, Switzerland

* pascale.gerbault@unige.ch

## Abstract

Yaws affects children in tropical regions, while syphilis primarily affects sexually active adults worldwide. Despite various campaigns towards the eradication of yaws and elimination of syphilis, these two diseases are still present in Ghana. The aetiological agents of both diseases, two *Treponema pallidum* subspecies, are genetically similar. This study aimed to assess the prevalence of these treponematoses and the occurrence of pathogens causing similar skin lesions in the Ashanti region of Ghana. A point-of-care test was used to determine the seroprevalence of the treponematoses. Both yaws and syphilis were identified in the Ashanti region of Ghana. Multiplex PCR was used to identify treponemes and other pathogens that cause similar skin lesions. The results indicated that the seroprevalences of *T. pallidum* in individuals with yaws-like and syphilis-like lesions were 17.2% and 10.8%, respectively. Multiplex PCR results showed that 9.1%, 1.8% and 0.9% of yaws-like lesions were positive for *Haemophilus ducreyi*, herpes simplex virus-1 (HSV-1) and *T. pallidum* respectively. Among syphilis-like lesions, 28.3% were positive for herpes simplex virus -2 (HSV-2) by PCR. To our knowledge, this is the first time HSV-I and HSV-2 have been reported from yaws-like and syphilis-like lesions, respectively, in Ghana. The presence of other organisms apart from *T. pallidum* in yaws-like and syphilis-like lesions could impede the total healing of these lesions and the full recovery of patients. This may complicate efforts to achieve yaws eradication by 2030 and the elimination of syphilis and warrants updated empirical treatment guidelines for skin ulcer diseases.

**Data Availability Statement:** All relevant data are within the manuscript and its Supporting Information files.

**Funding:** Global Research Challenge Fund - University of Westminster grant number SRO03150 (PG, SR) Bill and Melinda Gates Foundation grant number INV-036560 (JBP) Royal Society International Exchanges 2021 Round 3 award number IES\R3\213065 (SR, PPB) None of the sponsors or funders played any role in the study design, data collection and analysis, decision to publish, or preparation of the manuscript.

**Competing interests:** JBP reports research support from Gilead Sciences, non-financial support from Abbott Laboratories, and consulting for Zymeron Corporation, all outside the scope of this work. This does not alter our adherence to PLOS ONE policies on sharing data and materials. All other authors declare no com-peting interests.

## Introduction

Yaws is a neglected tropical disease which mostly affects the skin of children ≤15 years of age [1]. Yaws can be treated with a single oral course of azithromycin, which motivated the Morges strategy to have yaws eradicated by 2020 [2]. However, in 2021 more than 120,000 cases were reported from 13 countries, and renewed efforts are needed to achieve eradication [3, 4]. Interestingly, the agent causing yaws, *Treponema pallidum* subsp. *pertenue* (TPE), is genetically very similar to the agent causing syphilis, *Treponema pallidum* subsp. *pallidum* (TPA) [5, 6]. The most striking differences between these two diseases are their mode of transmission, the cohorts they affect and their geographical distribution. Yaws is transmitted in children by skin-to-skin contact [1] and occurs in warm and humid areas of the world [7]. Syphilis is a sexually or congenitally transmitted disease that affects adults or neonates and occurs worldwide [7]. *Treponema pallidum* subspecies remain challenging to grow under traditional laboratory conditions [8, 9], which is one of the reasons why our understanding of treponemal diversity and evolution is still limited. Nonetheless, developments in molecular biology approaches have enabled researchers to investigate patterns of genetic diversity and differentiation as well as the evolutionary history of *Treponema pallidum* [10–17]. These developments have notably enabled the genomic comparisons of strains causing yaws and syphilis [10, 11, 16], which facilitated the design of various multi-locus sequence typing (MLST) schemes to characterise strain diversity and examine whether strains carry one of the two known azithromycin resistance mutations [18, 19]. These approaches have importantly evidenced that intra-species and inter-species recombination can occur [13, 20–24]. This is especially relevant in areas where yaws is endemic and where strains causing yaws and syphilis can infect the same host.

Further aspects to be considered when tackling the eradication of yaws include inaccurate diagnoses and inadequate treatment. These may occur because other pathogens can cause skin lesions similar to treponemal lesions [25–27]. Among these pathogens are *Mycobacterium ulcerans*, causing buruli ulcers; *H. ducreyi*, causing chancroid; *Chlamydia trachomatis* causing lymphogranuloma venereum; and herpes simplex virus 1 (HSV-1) and herpes simplex virus 2 (HSV-2), causing genital and oral lesions [25–27].

In this study, we explored the prevalence of syphilis and yaws in the same geographical area, as well as the presence of three other pathogens known worldwide to be causing lesions similar to those of yaws and syphilis, i.e. HSV-1, HSV-2, *H. ducreyi*, and for which a multiplex PCR protocol exists [28]. Our findings are discussed further in the context of yaws and syphilis distribution and in relation to aspects that can affect the control management of yaws and syphilis, as well as other related diseases.

## Methods

### Ethical clearance

The study protocol was reviewed and approved by ethical review committees of the Komfo Anokye Teaching Hospital, Ghana (approval number: KATH-IRB/AP/048/20) and the School of Life Sciences, University of Westminster, London, England (approval number: ETH2021-0285).

### Inclusivity in global research

Additional information regarding the ethical, cultural, and scientific considerations specific to inclusivity in global research is included in the Supporting Information (S1 File).

## Study communities

The study areas for both yaws and syphilis were in the Ashanti region of Ghana (Fig 1). Participants with yaws-like lesions were recruited from three locations: Afigya Kwabre South municipality (Afigya), with study sites Adweratia and Mpobi ; Juaben municipality, with study sites Kotei and Atia ; and Atwima Kwanwoma municipality (Atwima), with study sites Brofoyedu and Bebu. Participants with syphilis-like lesions were recruited from two government hospitals: the Komfo Anokye teaching hospital (KATH) and South Suntreso hospital (SSH), both located in Kumasi metropolis (Fig 1).

## Sample and data collection

Participants were recruited from the study sites (hospitals and rural communities) from February to December 2021. All participants were explained the purpose of the study using a participant information sheet and gave their written informed consent to take part in this study. Since yaws affects mainly children (≤15 years of age with reported peak incidence occurring in children aged 6–10 years) living in deprived rural areas, we enrolled children from six deprived communities in the Ashanti region. Participants from 5–17 years of age gave their verbal assent before their guardians were asked to sign the informed consent form.

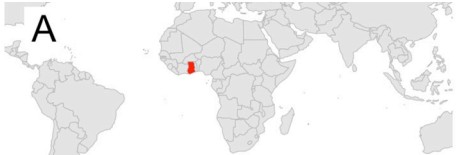

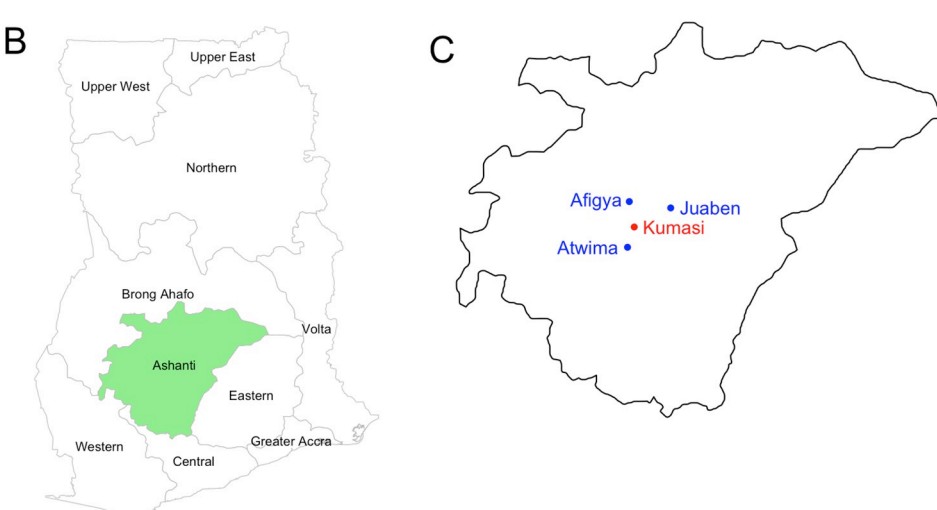

**Fig 1. Geographical location of the sites where participants with yaws-like and syphilis-like lesions were recruited.** A: Yaws endemic country where this study took place (Ghana, in red) B: Study area (the Ashanti region of Ghana, in light green) and C: Study sites in the Ashanti region of Ghana. Participants with yaws-like lesions were recruited from three locations (blue dots): Afigya Kwabre South municipality (Afigya), which had two study sites, Adweratia and Mpobi; Juaben municipality, which had two study sites, Kotei and Atia; and Atwima Kwanwoma municipality (Atwima), which had two study sites, Brofoyedu and Bebu. Participants with syphilis-like lesions were sampled from Kumasi metropolis (red dot), which had two study sites, KATH and SSH.

Participants who were 5 to 17 years of age with a minimum of one clinical lesion consistent with primary or secondary treponemal (yaws) infection [2] were enrolled for yaws-like lesion samples.

Individuals commonly affected with syphilis are young adults and adults and, in Ghana, they usually report to hospitals or pharmacies. This is why two government hospitals were visited for recruiting adult participants with syphilis-like lesions. Participants who were 18 years and above with a minimum of one clinical lesion consistent with primary or secondary treponemal infection (syphilis) were enrolled for syphilis-like lesion samples.

Age and sex were recorded for each participant. Additionally, participants were asked to self-define their ethnicity. However, no information related to sexual behaviour or activity was recorded, since this is beyond the scope of the present research. Individual participants remained anonymous, i.e. the authors have no access to information that could identify individual participants during or after data collection.

## Clinical diagnosis and seroprevalence

Clinical diagnoses of yaws- and syphilis-like lesions by study clinician and a trained community health nurse was established by identification of at least one skin lesion consistent with primary or secondary treponemal infection [2]. To determine the seroprevalence of treponematoses, the Chembio Dual Path Platform (DPP) screen and confirm test (Chembio Diagnostic Systems Inc. Medford, New York) was used according to the manufacturer's guidelines. This point-of-care test determines the presence of both treponemal and non-treponemal antibodies on a single platform within 15 minutes [29, 30].

Seropositivity for either treponematose was determined when both treponemal and non-treponemal antibody test lines were observed in the point-of-care test. This applies to both yaws-like and syphilis-like participants. In other words, participants we refer to as "DPP-positive" in this study showed both the treponemal and the non-treponemal test lines on their point-of-care test. When the point-of-care test was positive, participants were offered a single dose of azithromycin (for children) or an injection of benzathine penicillin G (for adults) by the local health care staff, as recommended by WHO [2, 3].

## Multiplex PCR

To identify common pathogens that cause yaws-like and syphilis-like lesions, skin lesions of all participants were sampled using Dacron swabs according to a standard protocol [31]. Skin lesion samples were then stored at -20˚C at the Kumasi Centre for Collaborative Research in Kumasi (KCCR), where DNA was extracted using QIAmp DNA mini kits (Qiagen, Germany) according to the manufacturer's instructions. A multiplex-PCR protocol previously developed for routine clinical practices specifically targeting molecular identification of *T. pallidum*, HSV-1/ 2 and *H. ducreyi* was applied [28]. This multiplex PCR was performed at the University Hospital Zurich, Switzerland, with the Roche LightCycler 96 (Roche Diagnosis, Switzerland) targeting the 16S ribosomal RNA gene [28] for both *T. pallidum* and *H. ducreyi*, and glycoprotein B (*gB*) region of HSV-1 and -2 as described by Glatz and colleagues [28] with few alterations as outlined below. Briefly, 20 µL reactions consisted of 10 µL Takyon No Rox Probe MasterMix (Eurogentec, Belgium), 0.5 µM of each primer, 0.1 µM of each probe, 1 µL internal control and 5 µL of extracted DNA. Reaction conditions were activation of uracil-N- glycosylase at 50˚C for 2 minutes, pre-denaturation at 95˚C for 3 minutes, 45 cycles of denaturation at 95˚C for 10 seconds and annealing and amplification at 63˚C for 45 seconds. Cycle threshold (Ct) value less than 40 cycles was taken as positive.

## Statistical analysis

Prevalence in this study refers to the number of participants who were positive with the DPP test (seroprevalence) or whose lesion showed DNA amplification in the multiplex PCR divided either by the number of participants in an area over the study period or by the total number of participants over this period. The Fisher's exact test (when lowest sample size was below 5) or Chi-squared test (when lowest sample size was five or above) were used to compare categorical characteristics between groups. A Bonferroni correction was applied for pairwise comparisons between groups to account for multiple testing. Statistical analyses were performed with R (version 4.2.2) on R Studio version 2022.07.1 Build 554. Figures were produced with the ggplot2 package (version 3.4.0) and maps were obtained using additional R packages, including rworldmap, maps, raster, countrycode, viridis, scales, dplyr, cowplot. R scripts used to produce the manuscript's Figures, Supporting Information Figures and statistical analyses are included in the Supporting Information (S2 File).

## Results

### Study population and demographics

During the study period, 110 participants with yaws-like lesions were recruited from six rural communities, and 46 participants with syphilis-like lesions were recruited from two hospitals in an urban area (Table 1). The median age of participants with yaws-like lesions was 8 years (range 5–17 years old). There was no significant difference between the number of male (n = 65) and female (n = 45) participants with yaws-like lesions (Chi square = 3.6, p-value = 0.06). Participants recruited for syphilis-like lesions had a median age of 29 years (range 19–75 years). There was no significant difference between the number of male (n = 17) and female (n = 29) participants with syphilis-like lesions (Chi square = 3.1, p-value = 0.08).

Participants in this study belonged to six groups, namely the Ashantis (the dominant ethnic group in the study area), the Akwapims, the Ewes, the Fantis, Akyim and Northerners, the latter referring to the Upper East, Upper West and Northern regions (Fig 2). Most participants

**Table 1. Seroprevalence of the treponematoses yaws and syphilis and number of lesions evidencing the occurrence of pathogens targeted by multiplex-PCR, according to study sites.**

| Study sites | Lesion types | N | DPP. negative | DPP. positive | seroprevalence | Multiplex PCR | | | | |
| --- | --- | --- | --- | --- | --- | --- | --- | --- | --- | --- |
| | | | | | | *T.pallidum* | *H.ducreyi* | HSV1 | HSV2 | PCR.negative |
| Adweratia | yaws-like | 65 | 54 | 11 | 16.9 | 0 | 10 | 0 | 0 | 55 |
| Mpobi | yaws-like | 23 | 16 | 7 | 30.4 | 0 | 0 | 1 | 0 | 22 |
| Bebu | yaws-like | 9 | 8 | 1 | 11.1 | 0 | 0 | 1 | 0 | 8 |
| Brofoyedu | yaws-like | 2 | 2 | 0 | 0 | 0 | 0 | 0 | 0 | 2 |
| Atia | yaws-like | 7 | 7 | 0 | 0 | 1 | 0 | 0 | 0 | 6 |
| Kotei | yaws-like | 4 | 4 | 0 | 0 | 0 | 0 | 0 | 0 | 4 |
| KATH | syphilis-like | 19 | 18 | 1 | 5.3 | 0 | 0 | 0 | 8 | 11 |
| SSH | syphilis-like | 27 | 23 | 4 | 14.8 | 0 | 0 | 0 | 5 | 22 |
| Total | | 156 | 132 | 24 | 15.4 | 1 | 10 | 2 | 13 | 130 |

Participants with yaws-like lesions were recruited from six study sites located in three regions: Adweratia and Mpobi in the Afigya Kwabre South municipality (Afigya on Fig 1C); Bebu and Brofoyedu located in Atwima Kwanwoma municipality (Atwima on Fig 1C); and Kotei and Atia located in Juaben municipality. Participants with syphilis-like lesions were recruited from two hospitals, KATH and SSH, located in Kumasi metropolis. N is the total number of lesions swabbed. DPP.negative and DPP. positive refer to the number of lesions that were serologically negative and serologically positive on the point-of-care test. Seroprevalence is the number of DPP-positive lesions divided by the total number of lesions swabbed in the corresponding study site. *T.pallidum*, *H.ducreyi*, HSV1, HSV2 refer to the number of lesions where the DNA from the corresponding pathogens was amplified with mulitplex-PCR. PCR.negative refers to the number of lesions where DNA was not amplified.

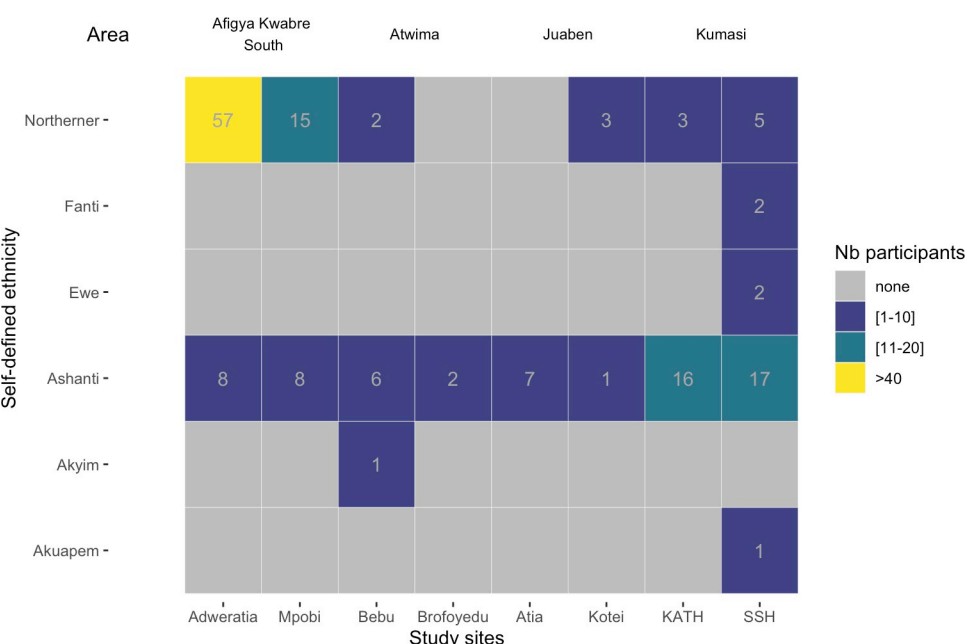

**Fig 2. Distribution of the participants self-defined ethnicity recruited for either yaws-like or syphilis-like lesions according to study site and municipality.** The colour scale shows the range of participants self-defined ethnicity recruited, where grey is not recruited, blue to green are low to medium numbers and above 40 is yellow. Participants recruited in this study belonged to six groups, namely the Ashantis (the dominant ethnic group in the study area), the Akwapims, the Ewes, the Fantis, Akyim and Northerners. Most participants were either Northerners (54.5%) or Ashantis (41.7%). The group that was disproportionately recruited were the Northerners. Northerner here refers to participants originating from the Upper East, Upper West and Northern regions (Fig 1B). This highlights that even within a region like the Ashanti region, the distribution of treponematoses is not homogeneous and specific investigations need to be performed in communities for yaws to be eradicated and syphilis to be eliminated (see S3 File for further details).

were either Northerners (54.5%) or Ashantis (41.7%). When yaws-like lesions were considered alone there were significantly more Northerners (77) than Ashantis (32) (Bonferroni adjusted p-value = $9 \times 10^{-5}$) or Akyim (1) recruited (Bonferroni adjusted p-value = $2 \times 10^{-16}$). However, when syphilis-like lesions were considered alone there were significantly more Ashantis (32) than Northerners (8) recruited (Bonferroni adjusted p-value = 0.00094).

## Distribution of lesions on body parts and prevalence of yaws and syphilis

The number of lesions that were consistent with primary and secondary yaws were 79 (71.8%) and 31 (28.2%), respectively, and those consistent with primary and secondary syphilis were 41 (89.1%) and five (10.9%), respectively. Lesions were spread on different parts of the body (Table 2). We found 75 participants (68%) with yaws-like lesions on the legs only or legs in combination with either one or two other locations such as the arm, belly, and skull (Table 2). There were 41 participants (37.3%) who had lesions on arms only or in combination with other locations, such as belly, buttocks, and skull. Most of the syphilis-like lesions (42 [91.3%]) were found in the genital area.

When combining participants with skin lesions consistent with either treponematose, the observed treponematose seroprevalence among study participants in the Ashanti region was 24/156 (15.4%). When yaws-like and syphilis-like lesions were considered separately, the observed seroprevalence among study participants was 19/110 (17.3%) and 5/46 (10.8%), respectively (Table 1). Among the study sites where yaws-like lesions were recorded, Mpobi

**Table 2. Distribution of skin lesions associated to yaws-like or syphilis-like lesions on the sampled body parts.**

| Location of lesions | Yaws-like lesions (n = 110) | Syphilis-like lesions (n = 46) |
|---|---|---|
| **Genital** | 0 | 42 |
| **Arms only** | 10 | 0 |
| **Arms and legs** | 28 | 0 |
| **Arms and belly** | 0 | 1 |
| **Arms, legs and belly** | 2 | 0 |
| **Arms, legs and skull** | 1 | 0 |
| **Arms, Buttocks and Belly** | 0 | 2 |
| **Belly only** | 3 | 0 |
| **Belly and chest** | 1 | 0 |
| **Buttocks only** | 4 | 0 |
| **Chin** | 2 | 0 |
| **Legs** | 44 | 0 |
| **Oral** | 7 | 1 |
| **Skull only** | 7 | 0 |
| **Ear** | 1 | 0 |

had the highest seroprevalence, followed by Adweratia and Bebu, with seroprevalences of 7/23 (30.4%), 11/65 (16.9%) and 1/9 (11.1%), respectively (Table 1).

Interestingly, the pairwise comparison between the seroprevalence of Adweratia (11/65; 16.9%) and Mpobi (7/23; 30.4%), located in the same district (Afigya Kwabre South district), did not show any significant difference (Bonferroni adjusted p-value = 0.2). However, the seroprevalence of Bebu (1/9; 11.1%), located in the Atwima Kwanwoma district, and of Mpobi (7/23; 30.4%) was significantly different (Bonferroni adjusted p-value = 0.008). Even though the seroprevalence of Adweratia (11/65; 16.9%) was higher than that of Bebu (1/9; 11.1%), the difference was not significant (Bonferroni adjusted p-value = 0.8). None of the participants recruited from the remaining three sites (Brofoyedu, Atia and Kotei) showed positive DPP tests. This suggests that yaws was more prevalent among study participants in the Afigya Kwabre South district (Adweratia and Mpobi) in the Ashanti region of Ghana compared to the other sites sampled. Even though the seroprevalence of syphilis at the SSH (4/27; 14.8%) was almost 3 times higher in comparison to that of KATH (1/19; 5.2%), this difference was not significant.

## Common microorganisms known to cause yaws-like and syphilis-like lesions identified by the multiplex polymerase chain reaction (PCR)

The multiplex real-time PCR confirmed the presence of *T. pallidum* and other common microorganisms that cause lesions like yaws and syphilis in 26 of the 156 lesions sampled. Amongst yaws-like lesions, 9% (10/110) were positive for *H. ducreyi*, 1.8% (2/110) for HSV-1 and 0.9% (1/110) for *T. pallidum*. The *H. ducreyi*- and the HSV-1-positive samples were associated with skin lesions of participants who had been DPP-positive or DPP-negative. The one sample that was *T. pallidum*-positive by multiplex PCR was identified from a yaws-like lesion of a participant who was DPP-negative, indicating an early stage of the disease.

Amongst syphilis-like lesions, 28.2% (13/46) were positive for HSV-2. HSV-2 was only identified in syphilis-like lesions (Table 3) while the DNA of *T. pallidum*, *H. ducreyi* or HSV-1 was not evidenced in any of the syphilis-like lesions sampled.

**Table 3. Number of lesions evidencing the occurrence of pathogens targeted by multiplex-PCR, according to the results of the point-of-care test.**

| Pathogen | DPP.negative | DPP.positive | nb | nb/Total (%) |
|---|---|---|---|---|
| *T. pallidum* | 1 | 0 | 1 | 0.6 |
| *H. ducreyi* | 8 | 2 | 10 | 6.4 |
| HSV1 | 1 | 1 | 2 | 1.3 |
| HSV2 | 13 | 0 | 13 | 8.3 |
| PCR negative | 109 | 21 | 130 | 83.3 |
| Total | 132 | 24 | 156 | 100 |

"nb" refers to the total number of lesions where the corresponding pathogen DNA (i.e. *T.pallidum*, *H.ducreyi*, HSV1, HSV2) targeted by mulitplex-PCR amplified. PCR.negative refers to lesions where DNA was not amplified. DPP. negative and DPP.positive refer to the number of lesions that were serologically negative and serologically positive, respectively, on the point-of-care test. "nb/Total" refers to the observed percentage of lesions evidencing the occurrence of the corresponding pathogen.

## Discussion

### Prevalence of yaws and syphilis in the Ashanti region of Ghana

This study used serological testing to assess the prevalence of treponematoses amongst participants with yaws-like or syphilis-like lesions in the Ashanti region of Ghana. The seroprevalences by DPP test in participants with yaws-like and syphilis-like lesions were 17.3% and 10.8%, respectively (Table 1). We compared this finding to other published studies that have reported the seroprevalence of yaws and syphilis in various countries (S1B and S1C Fig). This comparison highlighted that the seroprevalence of yaws and syphilis does not differ significantly globally or in yaws-endemic countries; even though it seems that significantly more studies on the prevalence of syphilis have been published than on the prevalence of yaws (see S3 File for extended discussion).

In addition to the serological test (DPP), multiplex PCR was performed for two main reasons. The first was to confirm serological testing results, i.e., it was expected that DPP-positive individuals would show the presence (amplification) of *T. pallidum* DNA from their lesions. We also anticipated that some of the DPP-negative samples might show the presence of *T. pallidum* DNA, since serological testing is known to miss the diagnosis of the very early stage of *T. pallidum* infection, when treponemes are in skin lesions, but not enough antibodies are present yet for detection by the DPP test [32]. Such was the case for one of the participants, whose yaws-like lesion was negative on the DPP serological test and multiplex-PCR-positive for *T. pallidum* (Table 1). Interestingly, none of the 24 lesions that showed a positive DPP test (serologically positive for *T. pallidum*), showed evidence of *T. pallidum* DNA with the multiplex PCR (Table 1). Such situations are often encountered in studies involving yaws-lesions sampling in the field [25, 27, 33]. As observed previously [25, 27], DPP positive individuals with negative *T. pallidum* DNA amplification may be in the latent stage of the disease, when they can be positive by serological testing (both treponemal and non-treponemal tests), but *T. pallidum* DNA may be absent or not present in enough quantity to be detected by the molecular approaches used here (multiplex PCR). The multiplex PCR identified *T. pallidum* in one yaws-like lesion but not in any of the syphilis-like lesions. Hence, if multiplex PCR had been the only means of assessing the prevalence of yaws and syphilis this estimated prevalence would have been 0.9% from yaws-like lesions (Table 1), or about 0.6% from all lesions sampled (Table 3).

The recommended routine diagnosis of active treponematoses (yaws and syphilis) is when an individual with skin lesions is tested serologically positive for both treponemal and non-

treponemal tests [2, 34]. The finding of 19 (17.2%) seropositive cases of *T. pallidum* and one (0.9%) PCR *T. pallidum* positive in yaws-like lesions here confirms that yaws is still present in these communities in the Ashanti region of Ghana, especially in the Afigya Kwabre South municipality (Adweratia and Mpobi), where 18 out of the 19 seropositive cases were found (Table 1). These two communities are adjacent in the Afigya Kwabre South municipality. Additionally, the group that was disproportionately recruited in this geographical region were the Northerners (Fig 2 and S3 File). This highlights that even within study sites, the distribution of yaws is not homogeneous and specific investigations need to be performed in communities for yaws to be eradicated.

The 10.8% *T. pallidum* seroprevalence among study participants with syphilis-like lesions confirms active or latent syphilis in these communities in Ghana. These infected participants, from a sexually active age range, could transmit the *T. pallidum* infection to an unborn baby, leading to congenital syphilis (S3 File).

This study confirms that both syphilis and yaws co-occur in the same geographical location (the Ashanti region of Ghana). Yaws and syphilis usually affect distinct cohorts (children and adolescents for yaws and adults for syphilis), which is one of the reasons why they have been studied and recorded independently. The results of this study emphasise the importance of investigating both diseases together in places where they co-occur, i.e. in all regions where yaws is endemic (see S1 Fig and S3 File for an extended discussion). It is unlikely to detect two *T. pallidum* subspecies DNA (e.g, yaws and syphilis treponemes) within a single lesion. However, *T. pallidum* can remain latent for years in the body after primary infection, implying an individual in a yaws-endemic region can acquire TPE (yaws-treponemes) as a child and TPA (syphilis-treponemes) as a young adult, leading to co-circulation of distinct treponeme subspecies within one host later in life. For example, in this study, the participant whose yaws-like lesion was positive for *T. pallidum* with the multiplex PCR was 17 years old, an age when this participant could also be sexually active and be infected with syphilis treponemes, leading to co-infection. Such a situation would enable recombination of yaws and syphilis treponemes.

Numerous studies [13, 20–24] have evidenced intergenomic recombination within *T. pallidum* subspecies as well as between species. How these recombination events occur remains to be established, but they imply that an individual can be co-infected by distinct strains of the same or of different treponeme subspecies. Variants resulting from recombination can pose diagnostic challenges because they can involve changes in conserved regions of DNA/RNA or proteins [13, 20–24], which tend to be the molecular targets of both serological and molecular diagnoses [24, 35] (see S3 File for extensive discussion on the impact of recombination).

False-negative diagnosis will negatively impact epidemiological data, which is essential to inform yaws eradication policies. In yaws-endemic countries, if for example a girl had yaws at childhood and undergoes antenatal-syphilis screening for syphilis when she is pregnant as an adult, she is likely to test positive by the treponemal test (the recommended test by WHO to diagnose syphilis in pregnancy in low income countries) even though she may not have syphilis. This is because this test can neither differentiate between past and current infections nor the treponemes that cause yaws and syphilis. This can have several negative effects, including unnecessary antibiotic administration, selection of antibiotic resistant strains and incorrect epidemiological data. Knowing both yaws and syphilis co-occur in the same geographical location is useful for guiding diagnosis (i.e, differentiating between past and current infections) and treatment (providing specific antibiotics and corresponding doses), and consequently eradication policies in theses areas. Studies investigating the presence of both diseases in a given region, and even within an individual, are very useful and can aid our understanding of strain diversification and evolution of *T. pallidum*.

### *H. ducreyi* and HSV-1 in yaws-like lesions

The second reason for applying multiplex PCR to all samples, regardless of their serological test results, was to identify other pathogens that could cause lesions similar to those of yaws and syphilis (including *H. ducreyi*, HSV-1 and HSV-2). The multiplex PCR results showed 9.1% (10/110) of the yaws-like lesions were positive for *H. ducreyi* and 1.8% (2/110) were positive for HSV-1 (Tables 1 and 3). The presence of *H. ducreyi* in yaws-like lesions in this study is consistent with previous studies on yaws [25, 26, 36–38]. Lesions of *H. ducreyi* are extremely difficult to differentiate clinically from those of yaws, and this bacterium can be found in individuals that are either seropositive or seronegative for yaws [27]. The results presented here are also consistent with this, as out of the 10 samples that were *H. ducreyi*-positive, two were seropositive for yaws and eight were seronegative.

Several studies have reported oral-facial [39–41] and genital infections [42–44] of HSV-1 worldwide but there seems to be a paucity of data of HSV-1 on skin sites other than the oral-facial and genital sites. In this study, HSV-1 was identified in two (1.8%) yaws-like lesions, including one oral lesion (lesion from a participant who was also DPP positive) and one lesion located on the arm. To our knowledge, this is the first time HSV-1 has been reported in yaws-like lesions.

### HSV-2 in syphilis-like lesions

Amongst the syphilis-like lesion samples, 13 (28.3%) were positive for HSV-2 (Tables 1 and 3). This is not surprising, considering HSV-2 has been reported as the leading cause of genital ulcers worldwide [45].

The presence of HSV-2 (Tables 1 and 3) in syphilis-like lesions evidenced in this study is a concern, as HSV-2 is known to enhance the spread of human immunodeficiency virus (HIV) by providing a favourable microenvironment that assists HIV infection in the host [46]. The dendritic cells (DCs) represent one of the first innate cell types that encounter HIV-1 and -2 in the genital mucosa. HSV-2 has been shown to modulate DCs, rendering them more receptive to HIV [46].

### The impact of pathogens other than *T. pallidum* on syphilis-like and yaws-like lesions

Routinely, *T. pallidum* infection diagnosis is based on serological testing and clinical signs. In this study, however, we noted three participants with yaws-like lesions who were serologically positive in the DPP-test, but the multiplex-PCR revealed other aetiologies instead of *T. pallidum*; i.e. in two cases *H. ducreyi* and in one case HSV-1. This suggests alternative causative agents of the current lesions, or more likely co-infection of the treponemal lesions with other pathogens. In low-and middle-income countries, such as Ghana, resources for molecular approaches such as PCR and sequencing are not readily available and most diagnosis methods for genital ulcers, like syphilis, are based on clinical examination and serological testing. Additionally, syndromic management of GUD is adopted in most of these low-middle-income countries [47, 48]. In syndromic management of GUD, all patients who present with genital ulcers receive a combined antibacterial and antiviral treatment targeting HSV-1 and -2, TPA and *H. ducreyi* together, without any further testing.

The low costs of syndromic management may outweigh that of specific treatment in low-middle-income countries, but this can lead to unnecessary drug use, possibly facilitating the emergence of antibiotic resistant strains. In addition, patients often find it challenging to fully comply with these combined antibacterial and antiviral drug regimens, and more specifically

in low-middle-income countries, not many patients can afford these combined treatments. These aspects may contribute to the persistence of these diseases and to the spread of other infections, such as HIV.

Notably, targeted diagnosis may be expensive, but it should reduce patient costs and limit antibiotic resistance. For example, a single dose of azithromycin is the recommended treatment against yaws and can be used to treat *H. ducreyi* infections [49], but it is unlikely that this same dose is effective to treat buruli ulcer [50]. Furthermore, Abdulai and colleagues, reported that yaws-like lesions caused by *H. ducreyi*, or other unknown pathogens, may continue to persist after a single round of mass treatment [27]. On the other hand, HSV infections will not respond to antibiotics. When lesions do not resolve after treatment, a wrong impression may be created among local communities leading health authorities think that yaws has not been eradicated from a previously endemic community, despite their joining the eradication campaign. This can result in negative perceptions of the yaws eradication programme. To achieve yaws eradication and the elimination of syphilis, total healing of ulcers is essential to improve the well-being of children and adults.

All these aspects argue for the need to invest in molecular approaches (PCR and sequencing) in yaws-endemic countries for diagnosing accurately and treating appropriately those affected, and to ensure total resolution of these lesions. Molecular approaches should target pathogens like *T pallidum*, *H. ducreyi*, HSV-1, *M. ulcerans* and probably *C. trachomonas* (serovars L1-L3). Knowing the precise aetiological agent (or agents) present in these lesions will have various positive impacts, including (i) improving disease management, (ii) providing the correct epidemiological data to (iii) inform health authorities and consequently disease elimination and eradication programs, and (iv) allowing targeting of groups or communities specifically affected.

## Limitations

This is an exploratory study, where the sample size per study site is relatively low ranging from 2 to 65 participants recruited. Despite some low sample sizes per location, when the observed seroprevalences of yaws and of syphilis were compared to those of other published studies that have reported similar data in various countries (S1B and S1C Fig), they fell within expected prevalence ranges for these diseases in Ghana and in other countries (S1A Fig). It should be stressed that the identification of yaws-like and syphilis-like lesions was based solely on clinical and epidemiological observation. The names of the lesions however are not used in our assessment of the prevalence of either treponematose and we do not think this affects our results. The molecular analysis of DNA was then performed as to provide a complementary examination of these lesions.

Other limitations include the data collected, including the number of lesions swabbed, where a single lesion per participant was swabbed, while swabbing distinct lesions may have increased the likelihood of *T. pallidum* DNA amplification. Furthermore, participant data recorded provided some information on the distribution of the treponematoses yaws and syphilis, but did not include behavioural data, such as sexual activity. This is however beyond the scope of the present study, which aimed at assessing the prevalence of yaws and syphilis within a single region in Ghana.

In order to assess whether other pathogens commonly found in skin lesions similar to those produced by *T. pallidum* subspecies could also be present in those skin lesions in Ghana, we applied a multiplex PCR protocol with high specificity and sensitivity established for the detection of worldwide transmitted pathogens in routine clinical practice [28]. However, DNA preservation in samples collected in so-called routine clinical practices cannot directly compare to

DNA preservation of the samples collected in the context of this study. One possibility for distinguishing between negative result and low quality sample would have been to add a control targeting human DNA in the multiplex PCR, as well as the inhibition control already integrated in this protocol. Note that various factors may have affected the preservation of the DNA in skin lesion samples after they were collected, including delay in courier service shipment of the samples from Ghana to the European continent and the placing of swabs in phosphate buffer saline (PBS) [51, 52]. Using specific DNA/RNA shield, which have been shown to preserve DNA better when temperature fluctuates [53], may be an alternative for improving DNA conservation in sampling situations such as those encountered in Ghana, where warm and humid conditions, as well as unstable refrigeration (due to instability of electricity supply), can affect the stability of DNA molecules.

Another limitation worth mentioning relates to the pathogens targeted with multiplex-PCR here. The multiplex-PCR protocol applied has been developed for routine clinical practice targeting only four of the pathogens known to cause lesions similar to those observed in cases of yaws and syphilis, i.e. *T. pallidum*, HSV-1/ 2 and *H. ducreyi* [28], while other pathogens, including *M. ulcerans* and *C. trachomatis*, can also be found in such lesions in Ghana. Our study emphasises the need to design sensitive and efficient multiplex-PCR protocols that could identify sets of pathogens found in specific types of lesions (for example pathogens causing skin ulcers).

In this context, this study focusses on the *T. pallidum* subspecies that have, to our knowledge, been reported in Ghana. However, lesions similar to yaws can also be observed in cases of bejel, another non-venereal treponematose caused by *T. p. endemicum*. *T. p. endemicum* has been reported in the Arabian Peninsula, more specifically in Iraq, Syria, and Saudi Arabia [54]. The African countries where *T. p. endemicum* has been reported include Mali, Niger, Burkina Faso and Senegal [55, 56]. The approaches we have used here do not allow to distinguish between *T. pallidum* subspecies and we cannot, at this stage, exclude that the lesions associated to yaws could actually be bejel lesions.

## Conclusion

This study was conducted in Ghana, one of the few countries where yaws is known to be endemic. To our knowledge (S2 Table, S3 and S4 Files), this study is the first to report the prevalence of yaws and syphilis together, from a single region (S1 Table). The prevalence of the treponematoses was assessed among 156 individuals showing lesions consistent with the diagnoses of syphilis and yaws. A seroprevalence of 17.3% (19/110) and 10.8% (5/46) of *T. pallidum* in individuals with yaws-like and syphilis-like lesions, respectively, indicate the occurence of treponemal strains causing yaws and syphilis within the Ashanti region of Ghana. Other pathogens known to cause similar skin lesions were also identified. *H. ducreyi* was more prevalent in yaws-like lesions, while HSV-2 was more prevalent in syphilis-like lesions. This is the first study reporting the presence of HSV-1 in yaws-like lesions (2/110).

The presence of other organisms apart from *T. pallidum* in yaws-like and syphilis-like lesions could prevent the total healing of these lesions and full recovery of patients. This may negatively affect the goal of yaws eradication by 2030 [3] and elimination of congenital syphilis.

## Supporting information

**S1 Fig. Prevalence of yaws and syphilis estimated from studies published between 1991 and 2020.** (A) Prevalence of yaws and syphilis estimated from 89 peer-reviewed studies (number of positive cases for yaws or syphilis divided by the total number of participants per study).

Out of the 15 countries known to be endemic for yaws, 13 are represented in the selected peer-reviewed articles reporting the prevalence of yaws or syphilis (no data published between 1991 and 2020 on the prevalence of yaws and syphilis from Cameroon and Liberia that fitted our criteria). Only four countries showed estimates of the prevalence of yaws and syphilis, from independent studies, i.e. Ecuador, Ghana, Indonesia, Nigeria. For comparison, blue triangles show the seroprevalence of yaws and syphilis estimated in the Ashanti region of Ghana from this study. There is no significant difference between any of the estimated prevalence of yaws and syphilis (Wilcoxon Rank Sum test p-values are 0.77 between the published prevalence of yaws and syphilis (globally); 0.67 between the published prevalence of yaws and syphilis in Ecuador; 0.70 between the published prevalence of yaws and syphilis in Ghana; 1 between the published prevalence of yaws and syphilis in Indonesia; 0.50 between the published prevalence of yaws and syphilis in Nigeria. The Chi-squared test p-value is 0.44 between the observed seroprevalence of yaws and syphilis in the Ashanti region). (B) Median of the estimated prevalence of yaws by country in studies published between 1991 and 2020. The borders of the four countries where the prevalence of both yaws and syphilis have been reported (Ecuador, Ghana, Indonesia, Nigeria) are represented in red. (C) Median of the estimated prevalence of syphilis by country in studies published between 1991 and 2020. The borders of the four countries where the prevalence of both yaws and syphilis have been reported (Ecuador, Ghana, Indonesia, Nigeria) are represented in red.
(TIFF)

**S1 Table. Data collected from the 156 participants from the Ashanti region of Ghana, this study.**
(CSV)

**S2 Table. Data from 89 peer-reviewed articles reporting the prevalence of yaws and syphilis.**
(CSV)

**S1 File. Inclusivity in global research questionnaire.**
(PDF)

**S2 File. Rscript for reproducing data analyses and graphs, including maps.**
(R)

**S3 File. Extensive discussion with references.**
(PDF)

**S4 File. References for the 89 peer-reviewed research articles in S2 Table.**
(DOCX)

## Acknowledgments

We thank Dr Patrick Kimmitt, Kostas Kampourakis, Helene Zondag and one anonymous Reviewer for comments on earlier versions of this manuscript. We are also grateful to Nicole Zimmermann for her work on the multiplex-PCR.

## Author Contributions

**Conceptualization:** Yaw Agyekum Boaitey, Alex Owusu-Ofori, Pascale Gerbault.

**Data curation:** Yaw Agyekum Boaitey.

**Formal analysis:** Yaw Agyekum Boaitey, Natasha Arora, Philipp P. Bosshard.

**Funding acquisition:** Jonathan B. Parr, Pascale Gerbault.

**Investigation:** Yaw Agyekum Boaitey.

**Methodology:** Yaw Agyekum Boaitey.

**Project administration:** Alex Owusu-Ofori, Natasha Arora, Jonathan B. Parr, Pascale Gerbault.

**Resources:** Amarachukwu Anyogu, Farhang Aghakhanian, Philipp P. Bosshard, Saki Raheem.

**Software:** Yaw Agyekum Boaitey, Farhang Aghakhanian, Natasha Arora, Jonathan B. Parr, Pascale Gerbault.

**Supervision:** Alex Owusu-Ofori, Amarachukwu Anyogu, Philipp P. Bosshard, Saki Raheem, Pascale Gerbault.

**Visualization:** Yaw Agyekum Boaitey, Pascale Gerbault.

**Writing – original draft:** Yaw Agyekum Boaitey, Alex Owusu-Ofori, Amarachukwu Anyogu, Philipp P. Bosshard, Saki Raheem, Pascale Gerbault.

**Writing – review & editing:** Yaw Agyekum Boaitey, Alex Owusu-Ofori, Amarachukwu Anyogu, Natasha Arora, Jonathan B. Parr, Pascale Gerbault.

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
