## [Decision Letter · Decision Letter 0]

26 Dec 2023

PONE-D-23-37271Prevalence of yaws and syphilis in the Ashanti region of Ghana and occurrence of H. ducreyi, Herpes simplex virus 1 and Herpes simplex virus 2 in skin lesions associated with treponematoses.PLOS ONE

Dear Dr. Gerbault,

Thank you for submitting your manuscript to PLOS ONE. After careful consideration, we feel that it has merit but does not fully meet PLOS ONE’s publication criteria as it currently stands. Therefore, we invite you to submit a revised version of the manuscript that addresses the points raised during the review process.

The work sometimes seems to be a review on Treponema's  instead of reporting test results. The paper should therefor be shortened and become more focussed.

We look forward to receiving your revised manuscript.

Kind regards,

Sylvia Maria Bruisten, Ph.D

Academic Editor

PLOS ONE

[We thank the Global Research Challenge Fund - University of Westminster grant number SRO03150 (PG, SR) for funding this research. We also thank the Royal Society and the Bill and Melinda Gates Foundation for funding molecular work and training, respectively. We thank Dr Patrick Kimmitt for comments on earlier version of this manuscript and Kostas Kampourakis.

This work was supported, in part, by the Bill & Melinda Gates Foundation [INV-036560]. Under the grant conditions of the Foundation, a Creative Commons Attribution 4.0 Generic License has already been assigned to the Author Accepted Manuscript version that might arise from this submission.]

 [Global Research Challenge Fund - University of Westminster grant number SRO03150 (PG, SR)

Bill and Melinda Gates Foundation grant number INV-036560 (JBP)

Royal Society International Exchanges 2021 Round 3 award number IES\\R3\\213065 (SR, PPB)

None of the sponsors or funders played any role in the study design, data collection and analysis, decision to publish, or preparation of the manuscript.]

[JBP reports research support from Gilead Sciences, non-financial support from Abbott Laboratories, and consulting for Zymeron Corporation, all outside the scope of this work. All other authors declare no competing interests]. 

5. We note that Figure 1B, 1C and 2 in your submission contain [map/satellite] images which may be copyrighted. All PLOS content is published under the Creative Commons Attribution License (CC BY 4.0), which means that the manuscript, images, and Supporting Information files will be freely available online, and any third party is permitted to access, download, copy, distribute, and use these materials in any way, even commercially, with proper attribution. For these reasons, we cannot publish previously copyrighted maps or satellite images created using proprietary data, such as Google software (Google Maps, Street View, and Earth). For more information, see our copyright guidelines: http://journals.plos.org/plosone/s/licenses-and-copyright.

a. You may seek permission from the original copyright holder of Figure 1B, 1C and 2 to publish the content specifically under the CC BY 4.0 license.  

Additional Editor Comments:

Both reviewers have good comments with which I fully agree. 

Reviewers' comments:

Reviewer's Responses to Questions

**Comments to the Author**

1. Is the manuscript technically sound, and do the data support the conclusions?

Reviewer #1: Yes

Reviewer #2: Yes

2. Has the statistical analysis been performed appropriately and rigorously? 

Reviewer #1: Yes

Reviewer #2: N/A

3. Have the authors made all data underlying the findings in their manuscript fully available?

Reviewer #1: Yes

Reviewer #2: Yes

4. Is the manuscript presented in an intelligible fashion and written in standard English?

Reviewer #1: Yes

Reviewer #2: Yes

5. Review Comments to the Author

Reviewer #1: GENERAL COMMENTS:

The manuscript titled “Prevalence of yaws and syphilis in the Ashanti region of Ghana and occurrence of H.ducreyi, Herpes simplex virus 1 and Herpes simplex virus 2 in skin lesions associated with treponematoses” by Agyekum et al. presents important results regarding the prevalence of T. pallidum and occurrence of different pathogens in skin lesions that are associated with treponematoses. While the manuscript contains interesting data that has a significant value to the research field, the manuscript has a lot of parts that are too detailed. This makes it harder to follow and takes away from the main results and aim of the work. My general comments and specific suggestions can be found below.

General comments

- Subspecies should be in lower case “Treponema pallidum subsp. pallidum".

- The overall manuscript needs to be more concise.

- The use of references is overabundant and can be greatly reduced in number.

- What is the reason that Treponema pallidum subspecies endemicum is not mentioned at all in the manuscript? The endemic areas of both diseases are very close and lesions may also occur in bejel. Please explain.

Introduction

1. Figure 1 shows the results of a literature study as described in the figure legend. However, if considered relevant for the context of the results, this should be included in the method section and the figure moved to the results.

Material and Methods

1. Line 141 – “Participants with syphilis” instead of “Syphilis participants”.

2. Include information on participants with yaws-like lesions (as mentioned in legend of Figure 2).

3. Lines 159 – 160 – “Participants from 5-17 years” add “of age” or rephrase.

4. Line 161 – Remove ‘from’

5. Line 165- define clinical diagnoses, is this based on age and clinical manifestations only? Are the participants asked about their sexual behavior/activity?

6. Line 180 – add ‘by’ before ‘Glatz’

7. What additional data was collected from the participants? In the results data is presented on ethnicity. Please add in the method section.

Results

1. The information presented in figures 3, 4 and 5 can be summarized in one table including statistical results. Also the demographics on age and sex as described in the first paragraph can be included.

2. Line 217 – “Participants were asked to self-define their ethnicity.” Move this information to the method section.

3. After mentioning that the ethnicity data comes from self-defining, this does not need to be repeated throughout the manuscript.

4. If the confidence intervals are considered relevant, please add this information to the summarized table as suggested earlier or, if the authors disagree with the suggestion, add this to figure 3.

5. The paragraph on seroprevalence only show percentages, please add full numbers as well.

6. The pairwise analysis of seroprevalence per region may be easier to follow by using a heatmap-like table or figure in order for the readers to have an overview of what is being explained.

Discussion

1. The discussion section is too long (10 pages) and should be shortened to half its size.

2. The serological assay was positive in a low percentage of the total suspected lesions of yaws and syphilis. What was the cause of the lesions if not these pathogens?

3. Lines 354 – 361 – The paragraph on congenital syphilis may be removed. The manuscript provides important data on prevalences, but its relevance to the elimination of congenital syphilis can be mentioned in one sentence.

4. I agree that the co-occurrence of syphilis and yaws in the same geographical area is interesting. Paragraph 362 – 379 should include information on Treponema pallidum subsp endemicum.

5. Paragraph on recombination should be removed as this is outside of the scope of this manuscript.

6. The limitation section could be more extensive including limitations in sample size per location, limited behavioral data, the use of a multiplex PCR with possibly less sensitive results?

7. Looking at the seroprevalence, misdiagnoses of syphilis and yaws seem to be very common as other ulcer-causing pathogen were detected instead of T. pallidum. Only 17.2% of the yaws-like lesions and 10.8% of the syphilis-like lesions were positive by serological testing (and one yaws-like lesions by PCR). This would suggest that 82.8% and 89.2% are not yaws or syphilis and may thereby have been misdiagnosed. If that is correct then please rephrase 522 – 524 “The prevalence … lesions determined.” Change “co-occurrence” into “occurrence”.

Reviewer #2: The manuscript “Prevalence of yaws and syphilis in the Ashanti region of Ghana and occurrence of H.

ducreyi, Herpes simplex virus 1 and Herpes simplex virus 2 in skin lesions associated

with treponematoses” by Yaw Agyekum Boaitey and colleagues aims to show the importance of correct diagnosis of the causative agent of ulcer lesions. The results of this study show presence of multiple pathogens present in ulcers of patients in Ghana, causing similar symptoms, and the need for applying the right treatment to prevent overuse of antibiotics. However, this reviewer has several concerns.

1. Figure 1: The description is very long and would be better suited for Methods section, as it describes the process in which it has been built. In boxplot part of the figure, it also shows results of the study, but the figure is placed in the Introduction section. It is not needed to show the data on endemic countries in a figure form. I would suggest either removing the figure, moving it to the Supplementary materials or keeping only the boxplot part of the figure and moving it to the Results section. Also, add a reference to the Supplementary table into the text.

2. Lines 124 to 127: Why did authors select these three pathogens to be analysed in their study? On lines 120 to 123, they mention additional pathogens able to cause skin lesions similar to those of treponemes. What was the reason behind selection of the pathogens? This needs to be addressed.

3. In Methods, Sample and data collection: The authors need to describe the process of acquiring samples and diagnosis determination in a greater detail. Was there an age criterium when selecting the patients for each group (see lines 161 to 165)? Were there patients with suspected yaws and syphilis in both rural and urban locations, and were they included? If not, why? What were the major factors deciding the clinical diagnosis and separation into yaws-like or syphilis-like lesion groups? Especially in the samples included into syphilis-like lesion group, from patients that had lesions on arms and belly, which is location more typical for yaws lesions.

4. Why were samples divided into these two groups, if there are many other pathogens that cause lesions similar to ones caused by treponematoses, a point which had been raised by the authors in the Introduction?

5. Was the seropositivity based on the density of the treponemal line in DPP test or just by visual presence of the line? Also, please include the sentence from lines 337 to 338 into methods, to be clear about what samples were considered serologically positive.

6. Line 178 to 180: A brief description of the multiplex PCR is needed. What were the targets for each pathogen and conditions of the reaction? Also, since the authors claim that one aim of the study was to assess the prevalence of treponematoses and since there is a possibility of misdiagnosis when using only serology, why authors did not use a target for the PCR that could differentiate between the two treponemal subspecies and confirm less reliable clinical diagnosis? Serology and non-specific PCR will also not allow for detection of possible coinfection by both yaws and syphilis, which is speculated in the Discussion.

7. Why did the authors not use a control in the multiplex PCR, for example detection of the patients' DNA, to differentiate between negative result and low quality sample?

8. Figure 4, showing ethnicity of the study participants is not needed, or could perhaps be combined with Figure 2 or 3, to show the distribution of the ethnicities in relation to rural/urban locations.

9. Lines 437 to 442: Why do authors state that all the syphilis-like lesion samples with DNA amplification by multiplex PCR were positive for HSV-2, if the prevalence of HSV-2 in these samples was 28.3% (13/46)? Please clarify what is meant by this sentence.

10. Since authors call for the molecular approaches to be implemented in yaws-endemic countries to diagnose the cause of the lesions, could they discuss what would be the appropriate pathogens to test for? In this study, the authors were detecting treponemes, as well as H. ducreyi and HSV-1 and HSV-2. Is there other pathogens present in Ghana, that could cause the lesions, but were not tested for, or do authors think that the negativity of the multiplex PCR for the analysed pathogens was mostly caused by the bad sample quality, that could have resulted from factors they mention in Limitations?

Minor comments:

1. Line 48: Yaws does affect not only children between 2 and 15 years of age, please rephrase.

2. Please unify the style in which the treponemal subspecies are referred to throughout the text. Authors use Treponema pallidum subsp. Pertenue, T. pallidum subsp pertenue, T. p. pertenue and TPE (without previous reference) at different places (same for subspecies pallidum).

3. Please label the A, B and C parts of Figure 2 accordingly, as described in text.

4. Add reference to the Supplementary Data to the Results and describe the data included there.

6. PLOS authors have the option to publish the peer review history of their article (what does this mean?). If published, this will include your full peer review and any attached files.

Reviewer #1: **Yes: **Helene Zondag

Reviewer #2: No

---

## [Author Response · Author response to Decision Letter 0]

8 Mar 2024

PONE-D-23-37271- REBUTTAL LETTER - 

Prevalence of yaws and syphilis in the Ashanti region of Ghana and occurrence of H. ducreyi, Herpes simplex virus 1 and Herpes simplex virus 2 in skin lesions associated with treponematoses.

PLOS ONE

Dear PLOS ONE Academic Editor Dr Sylvia Maria Bruisten,

We thank you and the two reviewers who have provided comments on our manuscript for considering our manuscript PONE-D-23-37271 for publication in PLOS ONE. This is a rebuttal letter that responds to each point raised by the academic editor and both reviewers. These points have been left in black font below. Our responses to each of those points is in blue font. When we cite sections of our revised manuscript, the text appears in black and italic font.

Please ensure that your manuscript meets PLOS ONE's style requirements, including those for file naming. The PLOS ONE style templates can be found at  https://journals.plos.org/plosone/s/file?id=wjVg/PLOSOne_formatting_sample_main_body.pdf and  https://journals.plos.org/plosone/s/file?id=ba62/PLOSOne_formatting_sample_title_authors_affiliations.pdf. 

We have modified the style of our manuscript so that level 1 headings read in font size 18pt and level 2 headings read in font size 16pt. We have also changed the word “Figure” for “Fig” when referring to one of the manuscript’s Figures. Tables have been formatted according to the PLOS ONE style template available online. 

On the title page, author surnames are in small letters, apart from the first letter, which is a capital letter. Numbers referring to author’s affiliations are all in superscript. The corresponding author’s name has an asterisk superscript and their email address is provided, with their initials in parenthesis next to it. 

We have completed PLOS’ questionnaire on inclusivity in global research and have included this file in the Supporting Information (S1 File). In addition, a section “Inclusivity in global research” has been added in the Methods section with the sentence “Additional information regarding the ethical, cultural, and scientific considerations specific to inclusivity in global research is included in the Supporting Information (S1 File)”. 

3. Thank you for stating the following in the Acknowledgments Section of your manuscript:  [We thank the Global Research Challenge Fund - University of Westminster grant number SRO03150 (PG, SR) for funding this research. We also thank the Royal Society and the Bill and Melinda Gates Foundation for funding molecular work and training, respectively. We thank Dr Patrick Kimmitt for comments on earlier version of this manuscript and Kostas Kampourakis. This work was supported, in part, by the Bill & Melinda Gates Foundation [INV-036560]. Under the grant conditions of the Foundation, a Creative Commons Attribution 4.0 Generic License has already been assigned to the Author Accepted Manuscript version that might arise from this submission.] We note that you have provided funding information that is not currently declared in your Funding Statement. However, funding information should not appear in the Acknowledgments section or other areas of your manuscript. We will only publish funding information present in the Funding Statement section of the online submission form.  Please remove any funding-related text from the manuscript and let us know how you would like to update your Funding Statement. Currently, your Funding Statement reads as follows:   [Global Research Challenge Fund - University of Westminster grant number SRO03150 (PG, SR) Bill and Melinda Gates Foundation grant number INV-036560 (JBP) Royal Society International Exchanges 2021 Round 3 award number IES\\R3\\213065 (SR, PPB) None of the sponsors or funders played any role in the study design, data collection and analysis, decision to publish, or preparation of the manuscript.]

We have removed the funding information from the Acknowledgments Section, which now reads as follows: “We thank Dr Patrick Kimmitt, Kostas Kampourakis, Helene Zondag and one anonymous Reviewer for comments on earlier versions of this manuscript. We are also grateful to Nicole Zimmermann for her work on the multiplex-PCR.”

We would like to modify the funding information in the Funding Statement section online to read as follows: 

“YAB research was funded by a PhD studentship from the Global Research Challenge Fund - University of Westminster, grant number SRO03150 (PG, SR).

This work was supported, in part, by the Bill and Melinda Gates Foundation, grant number INV-036560 (JBP). Under the grant conditions of the Foundation, a Creative Commons Attribution 4.0 Generic License has already been assigned to the Author Accepted Manuscript version that might arise from this submission.

This work was also partly supported by a Royal Society International Exchanges 2021 Round 3, award number IES\\R3\\213065 (SR, PPB).

None of the sponsors or funders played any role in the study design, data collection and analysis, decision to publish, or preparation of the manuscript.”

4. Thank you for stating the following in the Competing Interests section:  [JBP reports research support from Gilead Sciences, non-financial support from Abbott Laboratories, and consulting for Zymeron Corporation, all outside the scope of this work. All other authors declare no competing interests]. 

We have removed the Competing interests section from our manuscript and we confirm that the competing interests declared does not alter our adherence to PLOS ONE policies on sharing data and materials. Please update our online submission form so that the Competing Interests statement reads as follows:

“JBP reports research support from Gilead Sciences, non-financial support from Abbott Laboratories, and consulting for Zymeron Corporation, all outside the scope of this work. This does not alter our adherence to PLOS ONE policies on sharing data and materials. All other authors declare no competing interests.”

5. We note that Figure 1B, 1C and 2 in your submission contain [map/satellite] images which may be copyrighted. All PLOS content is published under the Creative Commons Attribution License (CC BY 4.0), which means that the manuscript, images, and Supporting Information files will be freely available online, and any third party is permitted to access, download, copy, distribute, and use these materials in any way, even commercially, with proper attribution. For these reasons, we cannot publish previously copyrighted maps or satellite images created using proprietary data, such as Google software (Google Maps, Street View, and Earth). For more information, see our copyright guidelines: http://journals.plos.org/plosone/s/licenses-and-copyright.   We require you to either (1) present written permission from the copyright holder to publish these figures specifically under the CC BY 4.0 license, or (2) remove the figures from your submission:   a. You may seek permission from the original copyright holder of Figure 1B, 1C and 2 to publish the content specifically under the CC BY 4.0 license.     We recommend that you contact the original copyright holder with the Content Permission Form (http://journals.plos.org/plosone/s/file?id=7c09/content-permission-form.pdf) and the following text: “I request permission for the open-access journal PLOS ONE to publish XXX under the Creative Commons Attribution License (CCAL) CC BY 4.0 (http://creativecommons.org/licenses/by/4.0/). Please be aware that this license allows unrestricted use and distribution, even commercially, by third parties. Please reply and provide explicit written permission to publish XXX under a CC BY license and complete the attached form.”   Please upload the completed Content Permission Form or other proof of granted permissions as an ""Other"" file with your submission.   In the figure caption of the copyrighted figure, please include the following text: “Reprinted from [ref] under a CC BY license, with permission from [name of publisher], original copyright [original copyright year].”   b. If you are unable to obtain permission from the original copyright holder to publish these figures under the CC BY 4.0 license or if the copyright holder’s requirements are incompatible with the CC BY 4.0 license, please either i) remove the figure or ii) supply a replacement figure that complies with the CC BY 4.0 license. Please check copyright information on all replacement figures and update the figure caption with source information. If applicable, please specify in the figure caption text when a figure is similar but not identical to the original image and is therefore for illustrative purposes only. The following resources for replacing copyrighted map figures may be helpful:   USGS National Map Viewer (public domain): http://viewer.nationalmap.gov/viewer/ The Gateway to Astronaut Photography of Earth (public domain): http://eol.jsc.nasa.gov/sseop/clickmap/ Maps at the CIA (public domain): https://www.cia.gov/library/publications/the-world-factbook/index.html and https://www.cia.gov/library/publications/cia-maps-publications/index.html NASA Earth Observatory (public domain): http://earthobservatory.nasa.gov/ Landsat: http://landsat.visibleearth.nasa.gov/ USGS EROS (Earth Resources Observatory and Science (EROS) Center) (public domain): http://eros.usgs.gov/# Natural Earth (public domain): http://www.naturalearthdata.com/

This is a misunderstanding, we apologise, as this information was not clear from our initial description of the methods. The maps shown on Figures 1B, 1C and 2A, 2B and 2C were all produced using R packages, including ggplot2, rworldmap, maps, raster, countrycode, viridis, scales, dplyr, cowplot. Because we produced those maps, no copyright permission needs to be sought. We have edited one of the sentences in the Statistical analysis subsection of the Methods section (Statistical analysis, page 8, lines 170-174) of the revised manuscript, so that it now reads: 

“Figures were produced with the ggplot2 package (version 3.4.0) and maps were obtained using additional R packages, including rworldmap, maps, raster, countrycode, viridis, scales, dplyr, cowplot. R scripts used to produce the manuscript’s Figures, Supporting Information Figures and statistical analyses are included in the Supporting Information (S2 File). ”

We have added this file in the Supporting information file list and uploaded the file. 

A section called Supporting Information, listing additional files and Figures, has been added at the end of the manuscript pages 31- 32, lines 662-692. This reads as follows:

“Supporting Information

S1 Fig. Prevalence of yaws and syphilis estimated from studies published between 1991 and 2020. (A) Prevalence of yaws and syphilis estimated from 89 peer-reviewed studies (number of positive cases for yaws or syphilis divided by the total number of participants per study). Out of the 15 countries known to be endemic for yaws, 13 are represented in the selected peer-reviewed articles reporting the prevalence of yaws or syphilis (no data published between 1991 and 2020 on the prevalence of yaws and syphilis from Cameroon and Liberia that fitted our criteria). Only four countries showed estimates of the prevalence of yaws and syphilis, from independent studies, i.e. Ecuador, Ghana, Indonesia, Nigeria. For comparison, blue triangles show the seroprevalence of yaws and syphilis estimated in the Ashanti region of Ghana from this study. There is no significant difference between any of the estimated prevalence of yaws and syphilis (Wilcoxon Rank Sum test p-values are 0.77 between the published prevalence of yaws and syphilis (globally); 0.67 between the published prevalence of yaws and syphilis in Ecuador; 0.70 between the published prevalence of yaws and syphilis in Ghana; 1 between the published prevalence of yaws and syphilis in Indonesia; 0.50 between the published prevalence of yaws and syphilis in Nigeria. The Chi-squared test p-value is 0.44 between the observed seroprevalence of yaws and syphilis in the Ashanti region). (B) Median of the estimated prevalence of yaws by country in studies published between 1991 and 2020. The borders of the four countries where the prevalence of both yaws and syphilis have been reported (Ecuador, Ghana, Indonesia, Nigeria) are represented in red. (C) Median of the estimated prevalence of syphilis by country in studies published between 1991 and 2020. The borders of the four countries where the prevalence of both yaws and syphilis have been reported (Ecuador, Ghana, Indonesia, Nigeria) are represented in red.

S1 Table. Data collected from the 156 participants from the Ashanti region of Ghana, this study.

S2 Table. Data from 89 peer-reviewed articles reporting the prevalence of yaws and syphilis.

S1 File. Inclusivity in global research questionnaire.

S2 File. Rscript for reproducing data analyses and graphs, including maps.

S3 File. Extensive discussion with references.

S4 File. References for the 89 peer-reviewed research articles in S2 Table.”

 Additional Editor Comments:  Both reviewers have good comments with which I fully agree.   We thank both reviewers for their comments. We address them point by point in our response below. The Reviewers’ point reads in black font and our response in blue. Text cited from the revised manuscript appears in black and italic font.

 5. Review Comments to the Author  Reviewer #1: GENERAL COMMENTS: The manuscript titled “Prevalence of yaws and syphilis in the Ashanti region of Ghana and occurrence of H.ducreyi, Herpes simplex virus 1 and Herpes simplex virus 2 in skin lesions associated with treponematoses” by Agyekum et al. presents important results regarding the prevalence of T. pallidum and occurrence of different pathogens in skin lesions that are associated with treponematoses. While the manuscript contains interesting data that has a significant value to the research field, the manuscript has a lot of parts that are too detailed. This makes it harder to follow and takes away from the main results and aim of the work. My general comments and specific suggestions can be found below.  General comments - Subspecies should be in lower case “Treponema pallidum subsp. pallidum”. This has been addressed throughout th

---

## [Decision Letter · Decision Letter 1]

4 Apr 2024

PONE-D-23-37271R1Prevalence of yaws and syphilis in the Ashanti region of Ghana and occurrence of H. ducreyi, herpes simplex virus 1 and herpes simplex virus 2 in skin lesions associated with treponematoses.PLOS ONE

Dear Dr. Gerbault,

Thank you for submitting your manuscript to PLOS ONE. After careful consideration, we feel that it has merit but does not fully meet PLOS ONE’s publication criteria as it currently stands. Therefore, we invite you to submit a revised version of the manuscript that addresses the points raised during the review process.

You provided an extensive rebuttal in which almost all points of the reviewers were addressed well. As the second reviewer points out, some minor points should still be answered to further improve the manuscript.

We look forward to receiving your revised manuscript.

Kind regards,

Sylvia Maria Bruisten, Ph.D

Academic Editor

PLOS ONE

Journal Requirements:

Reviewers' comments:

Reviewer's Responses to Questions

**Comments to the Author**

1. If the authors have adequately addressed your comments raised in a previous round of review and you feel that this manuscript is now acceptable for publication, you may indicate that here to bypass the “Comments to the Author” section, enter your conflict of interest statement in the “Confidential to Editor” section, and submit your "Accept" recommendation.

Reviewer #1: All comments have been addressed

Reviewer #2: (No Response)

2. Is the manuscript technically sound, and do the data support the conclusions?

Reviewer #1: Yes

Reviewer #2: Yes

3. Has the statistical analysis been performed appropriately and rigorously? 

Reviewer #1: N/A

Reviewer #2: Yes

4. Have the authors made all data underlying the findings in their manuscript fully available?

Reviewer #1: Yes

Reviewer #2: Yes

5. Is the manuscript presented in an intelligible fashion and written in standard English?

Reviewer #1: Yes

Reviewer #2: Yes

6. Review Comments to the Author

Reviewer #1: The author addressed all comments providing arguments to adjust or compromise on the suggestions that were made.

Reviewer #2: The authors addressed majority of the concerns raised by this reviewer. However, this reviewer has several minor points.

Minor points:

In Fig. 2: Black text on a dark blue background is hard to read.

In the Limitation section of this study, the authors should clearly state that the evidence for yaws and syphilis is solely based in clinical and epidemiological data and not on the molecular analysis of DNA.

The section of supplementary discussion „T. pallidum infections and congenital syphilis“ is not directly related to the study presented here and does not bring any new information. Moreover, the section of supplementary discussion focusing on recombination is perhaps too long, since in this study the co-infection is just a speculation, as the authors presents no evidence of it. It would be enough to mention the possibility of recombination including recombination in the primer site(s).

7. PLOS authors have the option to publish the peer review history of their article (what does this mean?). If published, this will include your full peer review and any attached files.

Reviewer #1: No

Reviewer #2: No

---

## [Author Response · Author response to Decision Letter 1]

3 May 2024

PONE-D-23-37271R1 - REBUTTAL LETTER - 

Prevalence of yaws and syphilis in the Ashanti region of Ghana and occurrence of H. ducreyi, herpes simplex virus 1 and herpes simplex virus 2 in skin lesions associated with treponematoses.

PLOS ONE

Dear PLOS ONE Academic Editor Dr Sylvia Maria Bruisten,

We thank Reviewer#2 who raised additional points during the review process and you for your comments to further improve our manuscript PONE-D-23-37271R1 for publication in PLOS ONE. This rebuttal letter addresses these points, left in black font below. Our responses to each of those points is in blue font. When we cite sections of our revised manuscript, the text appears in black and italic font. When we cite line numbers, we refer to the revised versions (of the manuscript or the S3 File supplementary information) without Track changes.

Journal Requirements:

Please review your reference list to ensure that it is complete and correct. If you have cited papers that have been retracted, please include the rationale for doing so in the manuscript text, or remove these references and replace them with relevant current references. Any changes to the reference list should be mentioned in the rebuttal letter that accompanies your revised manuscript. If you need to cite a retracted article, indicate the article’s retracted status in the References list and also include a citation and full reference for the retraction notice. 

We realised that reference 31 [Rod James. Guidelines on microbiological wound swabbing. 2018;(September)] that was cited for sampling the lesions with Dacron swabs was not accessible any longer. We cite here instead Munson et al. (2019). We have replaced our previous reference 31 with this one: [Munson M, Creswell B, Kondobala K, Ganiwu B, Lomotey RD, Oppong P, Agyeman FO, Kotye N, Diwura M, Ako EP, Simpson SV. Optimising the use of molecular tools for the diagnosis of yaws. Transactions of The Royal Society of Tropical Medicine and Hygiene. 2019 Dec 1;113(12):776-80.]. This article shows amplification of DNA from yaws-like lesions after sampling of skin ulcers with dry Dacron swabs, as used in our study.

6. Review Comments to the Author

Reviewer #1: The author addressed all comments providing arguments to adjust or compromise on the suggestions that were made.

We thank Reviewer #1 for their understanding and appreciation of our amendments to the original version of the manuscript.

Reviewer #2: The authors addressed majority of the concerns raised by this reviewer. However, this reviewer has several minor points.

We thank Reviewer #2 for their adding further aspects of our manuscript that need clarifying. We believe we have addressed those points below.

Minor points:

 

In Fig. 2: Black text on a dark blue background is hard to read. This black font colour has now been changed for grey. We have also checked this Figure with PACE tool as well and it reads fine. The S2 File showing the Rscript for reproducing data analyses and graphs, including maps, has also been updated accordingly.

In the Limitation section of this study, the authors should clearly state that the evidence for yaws and syphilis is solely based in clinical and epidemiological data and not on the molecular analysis of DNA.

Following Reviewer #2’s suggestion, we have inserted three sentences addressing this in the Limitation section lines 443-448 of the revised manuscript and read as follows:

“It should be stressed that the identification of yaws-like and syphilis-like lesions was based solely on clinical and epidemiological observation. The names of the lesions however are not used in our assessment of the prevalence of either treponematose and we do not think this affects our results. The molecular analysis of DNA was then performed as to provide a complementary examination of these lesions.”

Indeed, we are not using the names of the diseases as diagnosis evidence for those diseases, i.e. we are not using clinical and epidemiological data to assess the prevalence of either treponematose. We are using the names of the diseases in association to what the skin lesions looked like, i.e. “yaws-like lesions” and “syphilis-like lesions”. This is because when a lesion is examined, based on clinical and epidemiological observation, lesions that have been caused by treponemes may actually look like lesions caused by other pathogens, including H. ducreyii or herpes simplex viruses 1 or 2. This is why we used a point-of-care test in order to identify the presence of treponeme antigens in those lesions. We further want to emphasise that molecular analysis of DNA from lesions can miss the diagnosis of a treponematose (later stage of disease or very early stage of disease when treponeme DNA quantity is low) - and equally serological diagnosis can miss diagnosing a treponematose (early stage of the disease). We therefore do not think that a diagnosis based on clinical and epidemiological observation of lesions alone or a diagnosis based on molecular analysis of lesions alone is better than the other. We rather want to use them both as complementary tools that inform on the possible pathogens present in skin lesions that look like yaws and syphilis lesions.

The section of supplementary discussion „T. pallidum infections and congenital syphilis“ is not directly related to the study presented here and does not bring any new information. Moreover, the section of supplementary discussion focusing on recombination is perhaps too long, since in this study the co-infection is just a speculation, as the authors presents no evidence of it. It would be enough to mention the possibility of recombination including recombination in the primer site(s).

We agree with Reviewer #2 that the paragraph on “T.pallidum infections and congenital syphilis” did not bring much to the debate as it read. However, even though we do not assess congenital syphilis in participants, we do not agree that congenital syphilis is not related to the study we present here. We have consequently reworded the section on “T. pallidum infections and congenital syphilis” in the supplementary information to clarify why we think it matters to consider syphilis, congenital syphilis and yaws together. We have added one more reference to support this opinion, reference number 7 in the supplementary information S3 File (Tsuboi M, Evans J, Davies EP, Rowley J, Korenromp EL, Clayton T, et al. Prevalence of syphilis among men who have sex with men: a global systematic review and meta-analysis from 2000-20. Lancet Glob Health. 2021 Aug;9(8):e1110-e1118). The corresponding paragraph lines 95-106 (S3 File) now reads as follows:

“The 10.8% T. pallidum seroprevalence among study participants with syphilis-like lesions confirms active or latent syphilis in Ghana (Table 1). In this study, the median age of participants with syphilis-like lesions was 29 with minimum age of 19 and maximum age of 75, a sexually active age range. These infected participants could transmit their T. pallidum infection to an unborn baby, leading to congenital syphilis. WHO considers congenital syphilis as a public health problem [6]. To eliminate congenital syphilis, treating pregnant women is not sufficient: syphilis must also be eliminated from the general population. In fact, reducing the global syphilis incidence by 90% between 2018 and 2030 is one of the four ambitious targets detailed in the Global Health Sector strategy addressing sexually transmitted diseases [7]. We argue supporting accurate diagnosis of syphilis and specific assessment of the presence of its causative treponeme, T. pallidum pallidum, and of related treponemes, such as T. pallidum pertenue causing yaws, in countries such as Ghana, will be essential to achieving this Global Health Sector strategy target.”

We have shortened the section on recombination in the supplementary information S3 File lines 108-133 deleting some of the details on the recombined strains, and making our speculative statement explicit. This section and this hypothesis - even speculative - are however important to develop here as it has never been explicitly eluded to before and it needs to be raised to be looked into further in future studies on treponematoses.

7. PLOS authors have the option to publish the peer review history of their article (what does this mean?). If published, this will include your full peer review and any attached files.

Do you want your identity to be public for this peer review? For information about this choice, including consent withdrawal, please see our Privacy Policy.

Reviewer #1: No

Reviewer #2: No

We registered as PACE users, uploaded our Fig 1 and Fig 2 and checked they could be easily read and appeared clearly. We have then downloaded Fig 1 and Fig 2 back and uploaded these versions onto PLOS ONE Editorial Manager. 

On the Track changes version of our revised manuscript, we have also made a few minor changes that are listed below:

deletion of extra spaces lines 92, 102, 216, 265;

clarification of what “seropositivity” refers to in this study by adding “antibody” and in the point-of-care test” line 137;

adding “the latter” to clarify that the “the Upper East, Upper West and Northern regions” refer to “the Northerners” line 204;

deletion of unnecessary “in” line 214 (Fig 2 caption);

inserting “respectively” line 281 in the caption of Table 3 to clarify “DPP.negative” and “DPP.positive” definitions;

updating the reference to Tables 1 and 3 line 314 instead of Fig 4 from an earlier version of the manuscript;

ensuring the prevalence of yaws and syphilis estimated from multiplex-PCR in the text matches the reported percentage from Table 3 (0.6% and not 0.67%);

there are various reasons why yaws and syphilis have been reported independently, we only cite one of them here, consequently we specify “one of the reasons why” line 332;

we have inserted a comma line 334 (after co-occur);

we prefer “different” instead of “distinct” line 347 in relation to treponeme subspecies;

an “s” was missing at the end of “diagnose” line 350;

the list of references has been updated line 372;

a “-“ has been added lines 391 and 408 to make it clear the “2” refers to HSV-2;

we prefer the use of “in addition” in the context of the text line 412 rather than “on the other side”; 

we think “notably” is more accurate than “noteworthy” line 417;

line 424 we have replaced “and let” by “leading”;

we realised we refer to DNA “preservation” and not “conservation” lines 460, 461, 465 and 468;

line 477 we have replaced “of designing” by “to design”.

We hope we have addressed the Editor’s and Reviewer #2’s comments appropriately and that our manuscript now meets PLOS ONE publishing criteria. 

Best regards,

Pascale Gerbault, on behalf of the co-authors.

---

## [Editor Report · Decision Letter 2]

7 May 2024

Prevalence of yaws and syphilis in the Ashanti region of Ghana and occurrence of H. ducreyi, herpes simplex virus 1 and herpes simplex virus 2 in skin lesions associated with treponematoses.

PONE-D-23-37271R2

Dear Dr. Gerbault,

We’re pleased to inform you that your manuscript has been judged scientifically suitable for publication and will be formally accepted for publication once it meets all outstanding technical requirements.

Kind regards,

Sylvia Maria Bruisten, Ph.D

Academic Editor

PLOS ONE

Additional Editor Comments (optional):

The authors have done an excellent job at answering all comments and adjusting and improving the manuscript. The manuscript is now much more coherent and to the point with removing parts, that were previously too elaborately discussed, to the supplementary data.
---

## [Editor Report · Acceptance letter]

10 May 2024

PONE-D-23-37271R2 

PLOS ONE

Dear Dr. Gerbault, 

I'm pleased to inform you that your manuscript has been deemed suitable for publication in PLOS ONE. Congratulations! Your manuscript is now being handed over to our production team.

Kind regards, 

on behalf of

Dr. Sylvia Maria Bruisten 

Academic Editor

PLOS ONE